# Regularizing Deep Neural Networks by Noise: Its Interpretation and Optimization

**Hyeonwoo Noh**     **Tackgeun You**     **Jonghwan Mun**     **Bohyung Han**
Dept. of Computer Science and Engineering, POSTECH, Korea
`{shgusdngogo,tackgeun.you,choco1916,bhhan}@postech.ac.kr`

## Abstract

Overfitting is one of the most critical challenges in deep neural networks, and there are various types of regularization methods to improve generalization performance. Injecting noises to hidden units during training, *e.g.,* dropout, is known as a successful regularizer, but it is still not clear enough why such training techniques work well in practice and how we can maximize their benefit in the presence of two conflicting objectives—optimizing to true data distribution and preventing overfitting by regularization. This paper addresses the above issues by 1) interpreting that the conventional training methods with regularization by noise injection optimize the lower bound of the true objective and 2) proposing a technique to achieve a tighter lower bound using multiple noise samples per training example in a stochastic gradient descent iteration. We demonstrate the effectiveness of our idea in several computer vision applications.

## 1   Introduction

Deep neural networks have been showing impressive performance in a variety of applications in multiple domains [2, 12, 20, 23, 26, 27, 28, 31, 35, 38]. Its great success comes from various factors including emergence of large-scale datasets, high-performance hardware support, new activation functions, and better optimization methods. Proper regularization is another critical reason for better generalization performance because deep neural networks are often over-parametrized and likely to suffer from overfitting problem. A common type of regularization is to inject noises during training procedure: adding or multiplying noise to hidden units of the neural networks, *e.g.*, dropout. This kind of technique is frequently adopted in many applications due to its simplicity, generality, and effectiveness.

Noise injection for training incurs a tradeoff between data fitting and model regularization, even though both objectives are important to improve performance of a model. Using more noise makes it harder for a model to fit data distribution while reducing noise weakens regularization effect. Since the level of noise directly affects the two terms in objective function, model fitting and regularization terms, it would be desirable to maintain proper noise levels during training or develop an effective training algorithm given a noise level.

Between these two potential directions, we are interested in the latter, more effective training. Within the standard stochastic gradient descent framework, we propose to facilitate optimization of deep neural networks with noise added for better regularization. Specifically, by regarding noise injected outputs of hidden units as *stochastic* activations, we interpret that the conventional training strategy optimizes the lower bound of the marginal likelihood over the hidden units whose values are sampled with a reparametrization trick [18].

Our algorithm is motivated by the importance weighted autoencoders [7], which are variational autoencoders trained for tighter variational lower bounds using more samples of stochastic variables

per training example in a stochastic gradient descent iteration. Our novel interpretation of noise injected hidden units as stochastic activations enables the lower bound analysis of [7] to be naturally applied to training deep neural networks with regularization by noise. It introduces the importance weighted stochastic gradient descent, a variant of the standard stochastic gradient descent, which employs multiple noise samples in an iteration for each training example. The proposed training strategy allows trained models to achieve good balance between model fitting and regularization. Although our method is general for various regularization techniques by noise, we mainly discuss its special form, dropout—one of the most famous methods for regularization by noise.

The main contribution of our paper is three-fold:

- We present that the conventional training with regularization by noise is equivalent to optimizing the lower bound of the marginal likelihood through a novel interpretation of noise injected hidden units as stochastic activations.

- We derive the importance weighted stochastic gradient descent for regularization by noise through the lower bound analysis.

- We demonstrate that the importance weighted stochastic gradient descent often improves performance of deep neural networks with dropout, a special form of regularization by noise.

The rest of the paper is organized as follows. Section 2 discusses prior works related to our approach. We describe our main idea and instantiation to dropout in Section 3 and 4, respectively. Section 5 analyzes experimental results on various applications and Section 6 makes our conclusion.

## 2   Related Work

Regularization by noise is a common technique to improve generalization performance of deep neural networks, and various implementations are available depending on network architectures and target applications. A well-known example is dropout [34], which randomly turns off a subset of hidden units of neural networks by multiplying noise sampled from a Bernoulli distribution.

In addition to the standard form of dropout, there exist several variations of dropout designed to further improve generalization performance. For example, Ba *et al.* [3] proposed adaptive dropout, where dropout rate is determined by another neural network dynamically. Li *et al.* [22] employ dropout with a multinormial distribution, instead of a Bernoulli distribution, which generates noise by selecting a subset of hidden units out of multiple subsets. Bulo *et al.* [6] improve dropout by reducing gap between training and inference procedure, where the output of dropout layers in inference stage is given by learning expected average of multiple dropouts. There are several related concepts to dropout, which can be categorized as regularization by noise. In [17, 37], noise is added to weights of neural networks, not to hidden states. Learning with stochastic depth [15] and stochastic ensemble learning [10] can also be regarded as noise injection techniques to weights or architecture. Our work is differentiated with the prior study in the sense that we improve generalization performance using better training objective while dropout and its variations rely on the original objective function.

Originally, dropout is proposed with interpretation as an extreme form of model ensemble [20, 34], and this intuition makes sense to explain good generalization performance of dropout. On the other hand, [36] views dropout as an adaptive regularizer for generalized linear models and [16] claims that dropout is effective to escape local optima for training deep neural networks. In addition, [9] uses dropout for estimating uncertainty based on Bayesian perspective. The proposed training algorithm is based on a novel interpretation of training with regularization by noise as training with latent variables. Such understanding is distinguishable from the existing views on dropout, and provides a probabilistic formulation to analyze dropout. A similar interpretation to our work is proposed in [24], but it focuses on reducing gap between training and inference steps of using dropout while our work proposes to use a novel training objective for better regularization.

Our goal is to formulate a stochastic model for regularization by noise and propose an effective training algorithm given a predefined noise level within a stochastic gradient descent framework. A closely related work is importance weighted autoencoder [7], which employs multiple samples weighted by importance to compute gradients and improve performance. This work shows that the importance weighted stochastic gradient descent method achieves a tighter lower-bound of the ideal marginal likelihood over latent variables than the variational lower bound. It also presents that the

bound becomes tighter as the number of samples for the latent variables increases. The importance weighted objective has been applied to various applications such as generative modeling [5, 7], training binary stochastic feed-forward networks [30] and training recurrent attention models [4]. This idea is extended to discrete latent variables in [25].

# 3 Proposed Method

This section describes the proposed importance weighted stochastic gradient descent using multiple samples in deep neural networks for regularization by noise.

## 3.1 Main Idea

The premise of our paper is that injecting noise into deterministic hidden units constructs stochastic hidden units. Noise injection during training obviously incurs stochastic behavior of the model and the optimizer. By defining deterministic hidden units with noise as stochastic hidden units, we can exploit well-defined probabilistic formulations to analyze the conventional training procedure and propose approaches for better optimization.

Suppose that a set of activations over all hidden units across all layers, $\mathbf{z}$, is given by

$$\mathbf{z} = g(\mathbf{h}_\phi(\mathbf{x}), \epsilon) \sim p_\phi(\mathbf{z}|\mathbf{x}), \tag{1}$$

where $\mathbf{h}_\phi(\mathbf{x})$ is a deterministic activations of hidden units for input $\mathbf{x}$ and model parameters $\phi$. A noise injection function $g(\cdot, \cdot)$ is given by addition or multiplication of activation and noise, where $\epsilon$ denotes noise sampled from a certain probability distribution such as Gaussian distribution. If this premise is applied to dropout, the noise $\epsilon$ means random selections of hidden units in a layer and the random variable $\mathbf{z}$ indicates the activation of the hidden layer given a specific sample of dropout.

Training a neural network with stochastic hidden units requires optimizing the marginal likelihood over the stochastic hidden units $\mathbf{z}$, which is given by

$$\mathcal{L}_{\text{marginal}} = \log \mathbb{E}_{p_\phi(\mathbf{z}|\mathbf{x})} \left[ p_\theta(\mathbf{y}|\mathbf{z}, \mathbf{x}) \right], \tag{2}$$

where $p_\theta(\mathbf{y}|\mathbf{z}, \mathbf{x})$ is an output probability of ground-truth $\mathbf{y}$ given input $\mathbf{x}$ and hidden units $\mathbf{z}$, and $\theta$ is the model parameter for the output prediction. Note that the expectation over training data $\mathbb{E}_{p(\mathbf{x},\mathbf{y})}$ outside the logarithm is omitted for notational simplicity.

For marginalization of stochastic hidden units constructed by noise, we employ the reparameterization trick proposed in [18]. Specifically, random variable $\mathbf{z}$ is replaced by Eq. (1) and the marginalization is performed over noise, which is given by

$$\mathcal{L}_{\text{marginal}} = \log \mathbb{E}_{p(\epsilon)} \left[ p_\theta(\mathbf{y}|g(\mathbf{h}_\phi(\mathbf{x}), \epsilon), \mathbf{x}) \right], \tag{3}$$

where $p(\epsilon)$ is the distribution of noise. Eq. (3) means that training a noise injected neural network requires optimizing the marginal likelihood over noise $\epsilon$.

## 3.2 Importance Weighted Stochastic Gradient Descent

We now describe how the marginal likelihood in Eq. (3) is optimized in a SGD (Stochastic Gradient Descent) framework and propose the IWSGD (Importance Weighted Stochastic Gradient Descent) method derived from the lower bound introduced by the SGD.

### 3.2.1 Objective

In practice, SGD estimates the marginal likelihood in Eq. (3) by taking expectation over multiple sets of noisy samples, where we computes a marginal log-likelihood for a finite number of noise samples in each set. Therefore, the real objective for SGD is as follows:

$$\mathcal{L}_{\text{marginal}} \approx \mathcal{L}_{\text{SGD}}(S) = \mathbb{E}_{p(\mathcal{E})} \left[ \log \frac{1}{S} \sum_{\epsilon \in \mathcal{E}} p_\theta(\mathbf{y}|g(\mathbf{h}_\phi(\mathbf{x}), \epsilon), \mathbf{x}) \right], \tag{4}$$

where $S$ is the number of noise samples for each training example and $\mathcal{E} = \{\epsilon_1, \epsilon_2, ..., \epsilon_S\}$ is a set of noises.

The main observation from Burda *et al.* [7] is that the SGD objective in Eq. (4) is the lower-bound of the marginal likelihood in Eq. (3), which is held by Jensen's inequality as

$$
\begin{aligned}
\mathcal{L}_{\text{SGD}}(S) &= \mathbb{E}_{p(\mathcal{E})} \left[ \log \frac{1}{S} \sum_{\epsilon \in \mathcal{E}} p_\theta(\mathbf{y}|g(\mathbf{h}_\phi(\mathbf{x}), \epsilon), \mathbf{x}) \right] \\
&\leq \log \mathbb{E}_{p(\mathcal{E})} \left[ \frac{1}{S} \sum_{\epsilon \in \mathcal{E}} p_\theta(\mathbf{y}|g(\mathbf{h}_\phi(\mathbf{x}), \epsilon), \mathbf{x}) \right] \\
&= \log \mathbb{E}_{p(\epsilon)} \left[ p_\theta(\mathbf{y}|g(\mathbf{h}_\phi(\mathbf{x}), \epsilon), \mathbf{x}) \right] \\
&= \mathcal{L}_{\text{marginal}},
\end{aligned}
\tag{5}
$$

where $\mathbb{E}_{p(\epsilon)}[f(\epsilon)] = \mathbb{E}_{p(\mathcal{E})}\left[\frac{1}{S}\sum_{\epsilon \in \mathcal{E}} f(\epsilon)\right]$ for an arbitrary function $f(\cdot)$ over $\epsilon$ if the cardinality of $\mathcal{E}$ is equal to $S$. This characteristic makes the number of noise samples $S$ directly related to the tightness of the lower-bound as

$$
\mathcal{L}_{\text{marginal}} \geq \mathcal{L}_{\text{SGD}}(S+1) \geq \mathcal{L}_{\text{SGD}}(S). \tag{6}
$$

Refer to [7] for the proof of Eq. (6).

Based on this observation, we propose to use $\mathcal{L}_{\text{SGD}}$ $(S > 1)$ as an objective of IWSGD. Note that the conventional training procedure for regularization by noise such as dropout [34] relies on the objective with $S = 1$ (Section 4). Thus, we show that using more samples achieves tighter lower-bound and that the optimization by IWSGD has great potential to improve accuracy by proper regularization.

### 3.2.2 Training

Training with IWSGD is achieved by computing the weighted average of gradients obtained from multiple noise samples $\epsilon$. This training strategy is based on the derivative of IWSGD objective with respect to the model parameters $\theta$ and $\phi$, which is given by

$$
\begin{aligned}
\nabla_{\theta,\phi} \mathcal{L}_{SGD}(S) &= \nabla_{\theta,\phi} \mathbb{E}_{p(\mathcal{E})} \left[ \log \frac{1}{S} \sum_{\epsilon \in \mathcal{E}} p_\theta(\mathbf{y}|g(\mathbf{h}_\phi(\mathbf{x}), \epsilon), \mathbf{x}) \right] \\
&= \mathbb{E}_{p(\mathcal{E})} \left[ \nabla_{\theta,\phi} \log \frac{1}{S} \sum_{\epsilon \in \mathcal{E}} p_\theta(\mathbf{y}|g(\mathbf{h}_\phi(\mathbf{x}), \epsilon), \mathbf{x}) \right] \\
&= \mathbb{E}_{p(\mathcal{E})} \left[ \frac{\nabla_{\theta,\phi} \sum_{\epsilon \in \mathcal{E}} p_\theta(\mathbf{y}|g(\mathbf{h}_\phi(\mathbf{x}), \epsilon), \mathbf{x})}{\sum_{\epsilon' \in \mathcal{E}} p_\theta(\mathbf{y}|g(\mathbf{h}_\phi(\mathbf{x}), \epsilon'), \mathbf{x})} \right] \\
&= \mathbb{E}_{p(\mathcal{E})} \left[ \sum_{\epsilon \in \mathcal{E}} w_\epsilon \nabla_{\theta,\phi} \log p_\theta(\mathbf{y}|g(\mathbf{h}_\phi(\mathbf{x}), \epsilon), \mathbf{x}) \right],
\end{aligned}
\tag{7}
$$

where $w_\epsilon$ denotes an importance weight with respect to sample noise $\epsilon$ and is given by

$$
w_\epsilon = \frac{p_\theta(\mathbf{y}|g(\mathbf{h}_\phi(\mathbf{x}), \epsilon), \mathbf{x})}{\sum_{\epsilon' \in \mathcal{E}} p_\theta(\mathbf{y}|g(\mathbf{h}_\phi(\mathbf{x}), \epsilon'), \mathbf{x})}. \tag{8}
$$

Note that the weight of each sample is equal to the normalized likelihood of the sample.

For training, we first draw a set of noise samples $\mathcal{E}$ and perform forward and backward propagation for each noise sample $\epsilon \in \mathcal{E}$ to compute likelihoods and corresponding gradients. Then, importance weights are computed by Eq. (8), and employed to compute the weighted average of gradients. Finally, we optimize the model by SGD with the importance weighted gradients.

### 3.2.3 Inference

Inference in the IWSGD is same as the standard dropout; input activations to each dropout layer are scaled based on dropout probability, rather than taking a subset of activations stochastically. Therefore, compared to the standard dropout, neither additional sampling nor computation is required during inference.

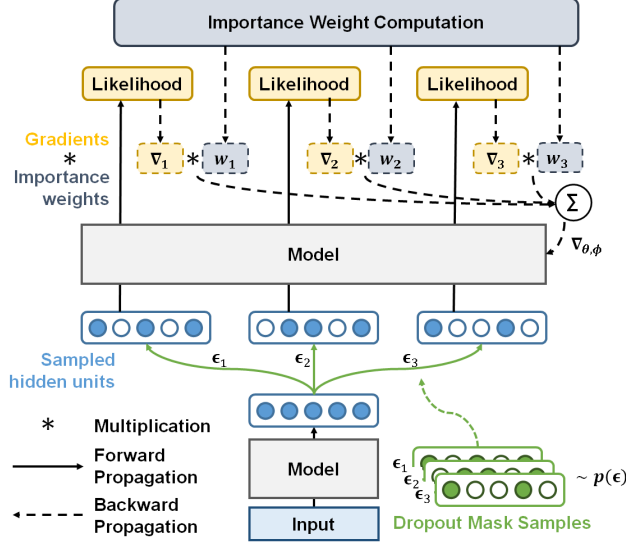

Figure 1: Implementation detail of IWSGD for dropout optimization. We compute a weighted average of the gradients from multiple dropout masks. For each training example the gradients for multiple dropout masks are independently computed and are averaged with importance weights in Eq. (8).

### 3.3 Discussion

One may argue that the use of multiple samples is equivalent to running multiple iterations either theoretically or empirically. It is difficult to derive the aggregated lower bounds of the marginal likelihood over multiple iterations since the model parameters are updated in every iteration. However, we observed that performance with a single sample is saturated easily and it is unlikely to achieve better accuracy with additional iterations than our algorithm based on IWSGD, as presented in Section 5.1.

## 4 Importance Weighted Stochastic Gradient Descent for Dropout

This section describes how the proposed idea is realized in the context of dropout, which is one of the most popular techniques for regularization by noise.

### 4.1 Analysis of Conventional Dropout

For training with dropout, binary dropout masks are sampled from a Bernoulli distribution. The hidden activations below dropout layers, denoted by $\mathbf{h}(\mathbf{x})$, are either kept or discarded by element-wise multiplication with a randomly sampled dropout mask $\epsilon$; activations after the dropout layers are denoted by $g(\mathbf{h}_\phi(\mathbf{x}), \epsilon)$. The objective of SGD optimization is obtained by averaging log-likelihoods, which is formally given by

$$\mathcal{L}_{\text{dropout}} = \mathbb{E}_{p(\epsilon)} \left[ \log p_\theta(\mathbf{y}|g(\mathbf{h}_\phi(\mathbf{x}), \epsilon), \mathbf{x}) \right], \tag{9}$$

where the outermost expectation over training data $\mathbb{E}_{p(\mathbf{x}, \mathbf{y})}$ is omitted for simplicity as mentioned earlier. Note that the objective in Eq. (9) is a special case of the objective of IWSGD with $S = 1$. This implies that the conventional dropout training optimizes the lower-bound of the ideal marginal likelihood, which is improved by increasing the number of dropout masks for each training example in an iteration.

### 4.2 Training Dropout with Tighter Lower-bound

Figure 1 illustrates how IWSGD is employed to train with dropout layers for regularization. Following the same training procedure described in Section 3.2.2, we sample multiple dropout masks as a realization of the multiple noise sampling.

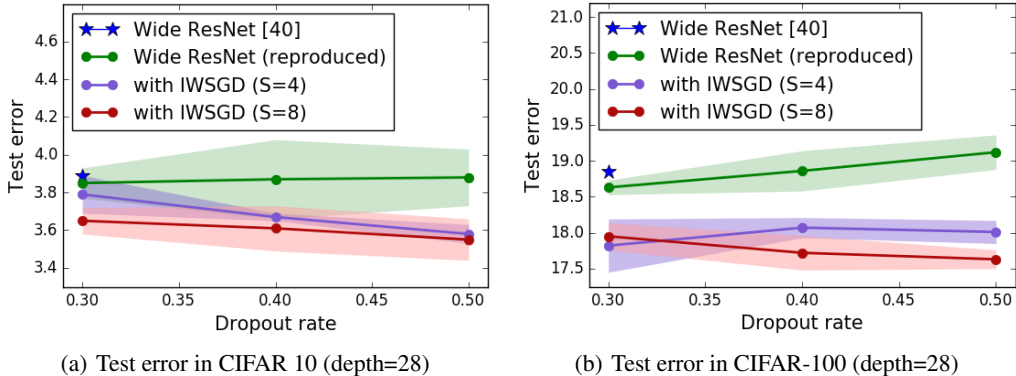

(a) Test error in CIFAR 10 (depth=28)        (b) Test error in CIFAR-100 (depth=28)

Figure 2: Impact of multi-sample training in CIFAR datasets with variable dropout rates. These results are with wide residual net (widening factor=10, depth=28). Each data point and error bar are computed from 3 trials with different seeds. The results show that using IWSGD with multiple samples consistently improves the performance and the results are not sensitive to dropout rates.

Table 1: Comparison with various models in CIFAR datasets. We achieve the near state-of-the-art performance by applying the multi-sample objective to wide residual network [40]. Note that ×4 iterations means a model trained with 4 times more iterations. The test errors of our implementations (including reproduction of [40]) are obtained from the results with 3 different seeds. The numbers within parentheses denote the standard deviations of test errors.

|  | CIFAR-10 | CIFAR-100 |
|---|---|---|
| ResNet [12] | 6.43 | - |
| ResNet with Stochastic Depth [15] | 4.91 | 24.58 |
| FractalNet with Dropout [21] | 4.60 | 23.73 |
| ResNet (pre-activation) [13] | 4.62 | 22.71 |
| PyramidNet [11] | 3.77 | 18.29 |
| Wide ResNet (depth=40) [40] | 3.80 | 18.30 |
| DenseNet [14] | 3.46 | 17.18 |
| Wide ResNet (depth=28, dropout=0.3) [40] | 3.89 | 18.85 |
| Wide ResNet (depth=28, dropout=0.5) (×4 iterations) | 4.48 (0.15) | 20.70 (0.19) |
| Wide ResNet (depth=28, dropout=0.5) (reproduced) | 3.88 (0.15) | 19.12 (0.24) |
| Wide ResNet (depth=28, dropout=0.5) with IWSGD ($S = 4$) | 3.58 (0.05) | 18.01 (0.16) |
| Wide ResNet (depth=28, dropout=0.5) with IWSGD ($S = 8$) | 3.55 (0.11) | 17.63 (0.13) |

The use of IWSGD for optimization requires only minor modifications in implementation. This is because the gradient computation part in the standard dropout is reusable. The gradient for the standard dropout is given by

$$\nabla_{\theta,\phi}\mathcal{L}_{\text{dropout}} = \mathbb{E}_{p(\epsilon)}\left[\nabla_{\theta,\phi}\log p_{\theta}\left(\mathbf{y}|g(\mathbf{h}_{\phi}(\mathbf{x}),\epsilon),\mathbf{x}\right)\right]. \qquad (10)$$

Note that this is actually unweighted version of the final line in Eq. (7). Therefore, the only additional component for IWSGD is about weighting gradients with importance weights. This property makes it easy to incorporate IWSGD into many applications with dropout.

# 5 Experiments

We evaluate the proposed training algorithm in various architectures for real world tasks including object recognition [40], visual question answering [39], image captioning [35] and action recognition [8]. These models are chosen for our experiments since they use dropouts actively for regularization. To isolate the effect of the proposed training method, we employ simple models without integrating heuristics for performance improvement (*e.g.*, model ensembles, multi-scaling, etc.) and make hyper-parameters (*e.g.*, type of optimizer, learning rate, batch size, etc.) fixed.

Table 2: Accuracy on VQA test-dev dataset. Our re-implementation of SAN [39] is used as baseline. Increasing the number of samples $S$ with IWSGD consistently improves performance.

|  | Open-Ended | | | | Multiple-Choice | | | |
|---|---|---|---|---|---|---|---|---|
|  | All | Y/N | Num | Others | All | Y/N | Num | Others |
| SAN [39] | 58.68 | 79.28 | 36.56 | 46.09 | - | - | - | - |
| SAN with 2-layer LSTM (reproduced) | 60.19 | 79.69 | 36.74 | 48.84 | 64.77 | 79.72 | 39.03 | 57.82 |
| with IWSGD ($S = 5$) | 60.31 | 80.74 | 34.70 | 48.66 | 65.01 | 80.73 | 36.36 | 58.05 |
| with IWSGD ($S = 8$) | 60.41 | 80.86 | 35.56 | 48.56 | 65.21 | 80.77 | 37.56 | 58.18 |

## 5.1 Object Recognition

The proposed algorithm is integrated into wide residual network [40], which uses dropout in every residual block, and evaluated on CIFAR datasets [19]. This network shows the accuracy close to the state-of-the-art performance in both CIFAR 10 and CIFAR 100 datasets with data augmentation. We use the publicly available implementation[1] by the authors of [40] and follow all the implementation details in the original paper.

Figure 2 presents the impact of IWSGD with multiple samples. We perform experiments using the wide residual network with widening factor 10 and depth 28. Each experiment is performed 3 times with different seeds in CIFAR datasets and test errors with corresponding standard deviations are reported. The baseline performance is from [40], and we also report the reproduced results by our implementation, which is denoted by Wide ResNet (reproduced). The result by the proposed algorithm is denoted by IWSGD together with the number of samples $S$.

Training with IWSGD with multiple samples clearly improves performance as illustrated in Figure 2. It also presents that, as the number of samples increases, the test errors decrease even more both on CIFAR-10 and CIFAR-100, regardless of the dropout rate. Another observation is that the results from the proposed multi-sample training strategy are not sensitive to dropout rates.

Using IWSGD with multiple samples to train the wide residual network enables us to achieve the near state-of-the-art performance on CIFAR datasets. As illustrated in Table 1, the accuracy of the model with $S = 8$ samples is very close to the state-of-the-art performance for CIFAR datasets, which is based on another architecture [14]. To illustrate the benefit of our algorithm compared to the strategy to simply increase the number of iterations, we evaluate the performance of the model trained with 4 times more iterations, which is denoted by $\times 4$ iterations. Note that the model with more iterations does not improve the performance as discussed in Section 3.3. We believe that the simple increase of the number of iterations is likely to overfit the trained model.

## 5.2 Visual Question Answering

Visual Question Answering (VQA) [2] is a task to answer a question about a given image. Input of this task is a pair of an image and a question, and output is an answer to the question. This task is typically formulated as a classification problem with multi-modal inputs [1, 29, 39].

To train models and run experiments, we use VQA dataset [2], which is commonly used for the evaluation of VQA algorithms. There are two different kinds of tasks: open-ended and multiple-choice task. The model predicts an answer for an open-ended task without knowing predefined set of candidate answers while selecting one of candidate answers in multiple-choice task. We evaluate the proposed training method using a baseline model, which is similar to [39] but has a single stack of attention layer. For question features, we employ a two-layer LSTM based on word embedding[2], while using activations from pool5 layer of VGG-16 [32] for image features.

Table 2 presents the results of our experiment for VQA. SAN with 2-layer LSTM denotes our baseline with the standard dropout. This method already outperforms the comparable model with spatial attention [39] possibly due to the use of a stronger question encoder, two-layer LSTM. When we evaluate performance of IWSGD with 5 and 8 samples, we observe consistent performance improvement of our algorithm with increase of the number of samples.

Table 3: Results on MSCOCO test dataset for image captioning. For BLEU metric, we use BLEU-4, which is computed based on 4-gram words, since the baseline method [35] reported BLEU-4 only.

|  | BLEU | METEOR | CIDEr |
|---|---|---|---|
| Google-NIC [35] | 27.7 | 23.7 | 85.5 |
| Google-NIC (reproduced) | 26.8 | 22.6 | 82.2 |
| with IWSGD ($S = 5$) | 27.5 | 22.9 | 83.6 |

Table 4: Average classification accuracy of compared algorithms over three splits on UCF-101 dataset. TwoStreamFusion (reproduced) denotes our reproduction based on the public source code.

| Method | UCF-101 |
|---|---|
| TwoStreamFusion [8] | 92.50 % |
| TwoStreamFusion (reproduced) | 92.49 % |
| with IWSGD ($S = 5$) | 92.73 % |
| with IWSGD ($S = 10$) | 92.69 % |
| with IWSGD ($S = 15$) | 92.72 % |

## 5.3 Image Captioning

Image captioning is a problem generating a natural language description given an image. This task is typically handled by an encoder-decoder network, where a CNN encoder transforms an input image into a feature vector and an LSTM decoder generates a caption from the feature by predicting words one by one. A dropout layer is located on top of the hidden state in LSTM decoder. To evaluate the proposed training method, we exploit a publicly available implementation[3] whose model is identical to the standard encoder-decoder model of [35], but uses VGG-16 [32] instead of GoogLeNet as a CNN encoder. We fix the parameters of VGG-16 network to follow the implementation of [35].

We use MSCOCO dataset for experiment, and evaluate models with several metrics (BLEU, METEOR and CIDEr) using the public MSCOCO caption evaluation tool. These metrics measure precision or recall of $n$-gram words between the generated captions and the ground-truths.

Table 3 summarizes the results on image captioning. Google-NIC is the reported scores in the original paper [35] while Google-NIC (reproduced) denotes the results of our reproduction. Our reproduction has slightly lower accuracy due to use of a different CNN encoder. IWSGD with 5 samples consistently improves performance in terms of all three metrics, which indicates our training method is also effective to learn LSTMs.

## 5.4 Action Recognition

Action recognition is a task recognizing a human action in videos. We employ a well-known benchmark of action classification, UCF-101 [33], for evaluation, which has 13,320 trimmed videos annotated with 101 action categories. The dataset has three splits for cross validation, and the final performance is calculated by the average accuracy of the three splits.

We employ a variation of two-stream CNN proposed by [8], which shows competitive performance on UCF-101. The network consists of three subnetworks: a spatial stream network for image, a temporal stream network for optical flow and a fusion network for combining the two-stream networks. We apply our IWSGD only to fine-tuning the fusion unit for training efficiency. Our implementation is based on the public source code[4]. Hyper-parameters such as dropout rate and learning rate scheduling is the same as the baseline model [8].

Table 4 illustrates performance improvement by integrating IWSGD but the overall tendency with increase of the number of samples is not consistent. We suspect that this is because the performance of the model is already saturated and there is no much room for improvement through fine-tuning only the fusion unit.

# 6 Conclusion

We proposed an optimization method for regularization by noise, especially for dropout, in deep neural networks. This method is based on a novel interpretation of noise injected deterministic hidden units as stochastic hidden ones. Using this interpretation, we proposed to use IWSGD (Importance Weighted Stochastic Gradient Descent), which achieves tighter lower bounds as the number of samples increases. We applied the proposed optimization method to dropout, a special case of the regularization by noise, and evaluated on various visual recognition tasks: image classification, visual question answering, image captioning and action classification. We observed the consistent improvement of our algorithm over all tasks, and achieved near state-of-the-art performance on CIFAR datasets through better optimization. We believe that the proposed method may improve many other deep neural network models with dropout layers.

**Acknowledgement** This work was supported by the IITP grant funded by the Korea government (MSIT) [2017-0-01778, Development of Explainable Human-level Deep Machine Learning Inference Framework; 2017-0-01780, The Technology Development for Event Recognition/Relational Reasoning and Learning Knowledge based System for Video Understanding].

## Footnotes

[1]https://github.com/szagoruyko/wide-residual-networks

[2]https://github.com/VT-vision-lab/VQA_LSTM_CNN

[3]https://github.com/karpathy/neuraltalk2

[4]http://www.robots.ox.ac.uk/~vgg/software/two_stream_action/

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
