[Reviews · NeurIPS 2017]

Reviewer 1



[UPDATE AFTER AUTHOR RESPONSE] I am happy with the authors' response. It's a valuable paper if the contributions are worked out more precisely and the relationship to Burda et al. [1] clarified. [ORIGINAL REVIEW] The authors propose a training procedure for networks with dropout or other forms of noise for regularization. Their approach is an importance weighted stochastic gradient descent algorithm that was originally proposed in the context of variational autoencoders by [1]. The contributions of the paper are (1) to use this approach in a new context (dropout) and (2) to show that it improves performance of a number of networks regularized with dropout. The paper is well written and straightforward to follow, and I think it is a valuable contribution. However, I do have a number of concerns that I will list in decreasing order of importance below: - Novelty. Unlike the authors’ repeated claims (e.g. lines 9–10, 41–44), their approach is not novel. Indeed, they do cite the original paper [1] repeatedly and even refer to results therein. Therefore, I suggest toning down the novelty claims a bit and working out more precisely the contribution of the paper. - Section 3.3. Why not run the test empirically and show the results? - Language. There are a couple of language mistakes, so please carefully proof-read the manuscript again. Some examples include line 108: *the* reparameterization trick; 156: *the* Bernoulli distribution; 164: within *each* mini-batch; 228: *the* MSCOCO dataset; 242f: *a* spatial stream, *a* temporal stream, *a* fusion network; 247: we observe *an* improvement. References: [1] Burda, Grosse & Salakhutdinov: Importance weighted auto encoders. ICLR 2016. https://arxiv.org/abs/1509.00519

Reviewer 2



This paper introduces a method for regularizing deep neural network by noise. The core of the approach is to draw connections between applying a random perturbation to layer activations and the optimization of a lower-bound objective function. Experiments for four visual tasks are carried out, and show a slight improvement of the proposed method compared to dropout. On the positive side: - The problem of regularization for training deep neural network is a crucial issue, which has a huge potential practical and theoretical impact. - The connection between regularization by noise and the derivation of a lower bound of the training objective is appealing to provide a formal understanding of the impact of noise regularization techniques, e.g. dropout. On the negative side: - The aforementioned connection between regularization by noise and training objective lower bounding seems to be a straightforward adaptation of [9] in the case of deep neural networks. For the most important result given in Eq (6), i.e. the fact that using several noise sampling operations gives a tighter bound on the objective function than using a single random sampling (as done in dropout), the authors refer to the derivation in [9]. Therefore, the improvement and positioning with respect to ref [9] is not clear to me. - Beyond this connection, the implementation of the model given in Section 4 for the case of dropout is also straightforward: several (random) forward passes are performed and a weighted average of the resulting gradients is computed using Eq (8), before applying the backward pass. - Experiments are carried out in various vision tasks, which choice could be better justified. It is not clear why these tasks are highly regularization demanding: the noise seems to be applied in a few (fully connected layers). In addition, the improvement brought out by the proposed method is contrasted across tasks (e.g. small improvement in UCF-101 and below the published baseline for image captioning). As a conclusion, although the approach is interesting, its novelty level seems to be low and the experiments should be more targeted to better illustrate the benefit of the multiple sampling operations. In their rebuttal, I would like the authors to answer my concern about positioning with respect to [9].

Reviewer 3



The paper interprets noise injection, especially dropout, in deep neural networks training as stochastic nodes in the network. This interpretation naturally leads to the observation that dropout, when trained with SGD, is a special case of training a stochastic model with one sample per mini-batch. The authors then proposed the increase the number of samples per mini-batch. Because dropout is usually applied to the top of the network, more samples lowers the variance of gradient estimation without increasing the computation cost by too much. This improves training efficiency. The authors performed experiments on multiple tasks including classification (cifar10), VQA, and image captioning. The proposed method consistently improved accuracy.